# OpenReview forum: "On Reward Maximization and Distribution Matching for Fine-Tuning Language Models"
_ICLR.cc/2022/Conference — ICLR 2022 Submitted_

### Official Review · Reviewer_qjvo · 2021-11-03

**Correctness:** 4
**Technical Novelty And Significance:** 3
**Empirical Novelty And Significance:** 3
**Recommendation:** 6
**Confidence:** 3

**Main Review:**

The idea to use a baseline to stabilize training in DM is rather simple but quite efficient as demonstrated by the experiments, and I think this is an overall useful, if somewhat incremental contribution. That being said is not quite clear to me that there isn't a simpler, more straightforward way to reduce variance in DM (see my full review for details).

Overall I think that the paper would be interesting to the community and I recommend acceptance, however I set my score as "borderline" until the concerns I raised in my review are addressed by the authors.

Strengths:
- Fine-tuning pre-trained language models with distributional preferences is an important research topic
- Simple, yet clever use of baselines.
- Convincing experimental results

Weaknesses:
- Some of the results of the paper (esp. section 3.2 on interpreting REINFORCE w/ KL control) are not particularly insightful as they don't really inform any modeling decision (as opposed to the "parametric reward" idea which suggests using variance reduction techniques from RL to apply to DM). What's more, I think that saying "KL-control developed in the RM paradigm can also be construed as belonging to DM" is slightly misleading because DM's core advantage is not so much in that it minimizes a KL but presumably that it minimizes the KL *in a specific direction*.
- Missing baseline for DM? It seems to me like optimizing the DM objective by actually sampling from the target distribution $p(x)$ (by eg. sampling from a large unlabeled dataset to get samples from $a$ and reweighting/subsampling according to $P(x)$) would be a more natural approach than first sampling from the model, and then reweighting with the likelihood ratio. The reason I think this is worth using as a baseline is that the variance issue the paper tries to address arises precisely because the DM objective is estimated by sampling from the model rather than sampling from the data. If we need RL variance reduction techniques solely because we are formulating a straightforward log-likelihood/cross-entropy objective as an RL objective, this begets the question: why formulate DM as RL in the first place when it doesn't need to be? I hope the authors can clarify this point.
- Results are reported exclusively as curves. I think that at least the core results should be reported as a table with a single number (with standard deviation) for each method (and each metric). It is also not entirely clear to me if the curves are training curves, or whether they represent quantities computed on a test/validation set.

**Summary Of The Paper:**

The paper builds upon previous work on fine-tuning pre-train language models to conform to pre-specified "distributional constraints" (such as: "the model should generate the same proportion of sentences with female or male subjects", etc...). Specifically, the authors draw parallel between a previous approach (distribution matching, DM) and RL-based approaches a la REINFORCE. Based on this comparison, they propose to adapt variance reduction techniques from RL (in particular the use of a baseline) to DM in order to facilitate training. Experimental results show that the resulting improved approach boasts better sample-efficiency, improved stability and overall competitive performance.

**Summary Of The Review:**

The idea to use a baseline to stabilize training in DM is rather simple but quite efficient as demonstrated by the experiments, and I think this is an overall useful, if somewhat incremental contribution. That being said is not quite clear to me that there isn't a simpler, more straightforward way to reduce variance in DM (see my full review for details).

Overall I think that the paper would be interesting to the community and I recommend acceptance, however I set my score as "borderline" until the concerns I raised in my review are addressed by the authors.

---

> ### Author Response · Authors · 2021-11-22
> **Reply to Reviewer qjvo**
>
> Thank you for appreciating the contributions of our work, recommending it for acceptance, for your insightful comments, and the interesting question about supervised training.
>
>
> > For eg. sampling from a large unlabeled dataset to get samples from $a$ and reweighting/subsampling according to $P(x)$) would be a more natural approach than first sampling from the model, and then reweighting with the likelihood ratio.
>
> To summarize, you are suggesting that instead of DPG one could do:
> - (1) offline sampling: produce a set of samples from $p$ by sampling from $a$ and reweighting with $P$
> - (2) supervised learning: of $\pi_\theta$ on this set of samples.
> The hypothesis here is that the CE objective of supervised learning might suffer less variance and will not need a variance reduction technique.
>
> Step 1 is equivalent to distilling a dataset using “Rejection Sampling” in particular (Parshakova et al-a 2019: “Global Autoregressive Models ...”) *did exactly that on low dimensional simulated data*. However, in high dimensional setups, there are several prohibitive issues with this approach:
>
> Rejection Sampling is only possible when one can use a proposal distribution $q$ s.t. the ratio $P(x)/q(x)$ is globally smaller than a fixed value $M$, which is not always possible (e.g. for distributional constraints), or even when possible, may lead to a very inefficient sampling process (a well-known issue of Rejection Sampling in high-dimensional spaces). This inefficiency will make it impossible to obtain a large enough dataset for supervised training.
>
> **DPG has two main advantages over such “offline sampling + supervised learning” approaches:**
> - (1) **Generality**: it can start from any proposal $q$,
> - (2) **The KL-DPG algorithm is adaptive** i.e. the proposal becomes closer and closer to the target $p$, which in turn makes the $q$-based sampler inside the algorithm more and more efficient. This adaptiveness is crucial, as was confirmed experimentally in (Khalifa et al, 2021, page 20 appendix), in particular in the case of EBMs for rare events. Without capitalizing on an evolving proposal, training becomes very inefficient e.g. for a single word control condition with 1/10^4 rarity the policy barely moves.
>
>
> > interpreting REINFORCE w/ KL control) are not particularly insightful as they don't really inform any modeling decision
> > DM's core advantage is not so much in that it minimizes a KL but presumably that it minimizes the KL in a specific direction.
>
>
> Thanks for raising this point, although we believe that reinterpreting KL control under the lens of DM is an important contribution for the following reasons:
> - (1) The similarities between the KL penalty (Ziegler et al. 19) and the EBM of minimal KL (Khalifa et al. 21) were not obvious and sparked hours of discussions, which even included the authors of (Ziegler et al.  19) themselves. We believe that our insights, crystallized as proof in the paper (eq 11) will benefit the community that is exploring such methods because it provides a new conceptual framework that was not available before.
>
> - (2) As you mentioned, indeed the target of DM is to minimize the KL in a specific direction. Currently the emergent EBM form in KL-control has no independent motivation. While this is not directly addressed in the current work, as we mention in the conclusion section, shaping an EBM of choice to be on the implicit EBM form of KL-control is an important future direction that can capitalize on strong RL algorithms such as PPO for optimizing that objective.
>
>
> > The core results should be reported as a table.
>
> Great suggestion, **we updated the manuscript to include a table for all the 10 tasks**  (please see page 25 in the appendix). Overall, the addition of a baseline (GDC++) outperforms GDC in 7/10 tasks in constraint satisfaction and 10/10 in terms of KL(p,pi) the distance from the target distribution (convergence)), this is indeed a better view to highlight the better performance of GDC++.
>
> > It is also not entirely clear to me if the curves are training curves or computed on a test/validation set.
>
> They are computed on a fresh batch of samples from $\pi_\theta$ at each validation step. We updated the paper to clarify this.
>
> > Overall I think that the paper would be interesting to the community and I recommend acceptance, however, I set my score as "borderline" until the concerns I raised in my review are addressed by the authors.
>
> Thank you for your insightful feedback on our work. We hope that the new draft and rebuttal address your concerns. If this is the case, we will appreciate it if you would increase your evaluation of our paper.

---

> > ### Comment · Reviewer_qjvo · 2021-12-02
> > **Response to rebuttal**
> >
> > I would like to thank the authors for their detailed response to my initial review. My apologies for the late follow up.
> >
> > I still think the paper makes a positive contribution and I am not as concerned as other reviewers about the perceived novelty of the approach: if it works, even if it is simple, it is still a useful contribution!
> >
> > That being said, I am still not entirely convinced by the theoretical novelty claimed by the authors.
> >
> > For instance, regarding the importance of reinterpreting REINFORCE + KL control as DM matching in a different KL direction, the authors state that "[their] insights, [...] will benefit the community that is exploring such methods because it provides a new conceptual framework that was not available before". I think introducing a new conceptual framework is only valuable insofar as it enables us to make useful decisions in terms of modeling that were not immediately apparent under previous frameworks. I don't think this is the case for that particular contribution in the context of this paper.
> >
> > Regarding the author's stated contribution of introducing "RL with parametric rewards", I would be a bit weary of making such a grand claim. Indeed, it seems to me that any log likelihood objective of the form $\mathbb  E_{x\sim q} \log p_\theta(x)$ can be reformulated as "RL with parametric rewards" by ways of importance reweighting $\mathbb E_{x\sim p_\theta} \frac{q(x)}{p_\theta(x)}\log p_\theta(x)$. Doesn't this make the definition somewhat vacuous?
> >
> > Overall while my evaluation of the paper still remains globally positive, I think these issues on the theoretical side somewhat undermine its core contribution (which is, as I previously said, valuable despite being simple). This is a difficult decision for me but i think I will keep my score to 6.

---

> ### Author Response · Authors · 2021-12-02
> **Feedback on our rebuttal?**
>
> Dear Reviewer qjvo, we were wondering whether we have successfully answered your questions and whether we have addressed your concerns with the changes included in the new version of the paper. We are looking forward to hearing your feedback.

---

### Official Review · Reviewer_Up1T · 2021-11-04

**Correctness:** 3
**Technical Novelty And Significance:** 2
**Empirical Novelty And Significance:** 2
**Recommendation:** 5
**Confidence:** 4

**Main Review:**

Strengths:

- presents novel baselines for DPG, which improve optimization performance.

Weaknesses:

- The connection between importance sampling based distillation and REINFORCE was introduced in in the DPG paper, and from this perspective, adding a baseline is a straightforward exercise, and lower in novelty.

- While baselines improve the DPG objective, constraint satisfaction, as shown by the second subfigure in figure 3, is NOT improved, contrary to claims in the abstract and main body of the paper.
- With lower objective and equal constraint satisfaction, text generated by $\pi$ may nevertheless be better. Are the improvements evident? No test results, quantitative or qualitative, are given on an application task to demonstrate this (the Khalifa 2021 paper, which utilizes the same technique less baselines, shows improved constraint satisfaction by using DPG-based DM, which is the point of utilizing DPG since EBMs are more difficult to sample)
- More generally, it is not clear to me that the DPG approach (baseline or not) is highly effective. The constraints in the EBM are NOT satisfied in either case, it would be great to investigate this in detail and figure out why.
- Related, the importance sampling step in DPG assumes that q(x)>0 where p(x)>0 to make the equality in (5) true, although this in the current context feels like a minor limitation.
- Based on Algorithm 1, it seems that the DPG importance weights are not normalized, which is the most common way to reduce variance (at the expense of bias) in IS. Is this the case?

**Summary Of The Paper:**

The authors explore the connections between reward maximimization (RM) with REINFORCE and distribution matching (DM) using distributional policy gradients (DPG), which uses importance sampling from a proposal distribution $q$ to minimize  $L = E_p[log\pi]$, where $p(x) = P(x)/Z$ is a distribution (energy-based model) that is difficult to sample from (but incorporates generation constraints, such as gender percentage, etc.). The corresponding DM sequence-level reward in REINFORCE is p/q. The authors derive and utilize $E_q[p/q] = \pi/q*Z$ as a baseline, which lowers variance, and leads to improvements in loss L and sampling efficiency.


**Summary Of The Review:**

Baselines for DPG methods are introduced, which improve optimization performance. However constraint satisfaction, the ultimate goal of the process, does not appear to improve, and neither qualitative nor qualitative evidence of improved test performance has been included. This, and the fact that DPG baselines are quite straightforward to work out, make the contribution of the paper seem quite low.

---

> ### Author Response · Authors · 2021-11-22
> **Reply to Reviewer Up1T**
>
> Thank you for your constructive feedback, we hereby tackle your concerns about “theoretical novelty” and constraint satisfaction of  DPG with and without baseline one by one:
>
> > adding a baseline is a straightforward exercise, and lower in novelty.
>
> The simple form of baselines ($B=Z$) that superficially resembles the derivation of baselines in standard Policy Gradients might give the impression that this is a trivial exercise and that our theoretical contributions in the paper are lower in novelty.
> We invite you to observe an important caveat that the reward term in the DPG case is **parametric** (i.e. rely on $\theta$) and disregarding this fact and applying RL baselines derivation for non-parametric rewards out of the box is not immediately motivated (for e.g. it could lead to biased gradients).
>
> For this, we had to introduce a generalization of the RM framework to the *parametric reward* case (an important core contribution of our work). Moreover, without a principled derivation, expanding this form to the DPG_off ($B_{off}(x) = Z \frac{\pi(x)}{q(x)}$) case wouldn't have been possible. For each, we do careful checks to verify unbiasedness and zero advantage (Appendix B).
>
> We clarify the presentation of our theoretical contributions in the paper **in a separate comment entitled ”theoretical novelty’’**.
>
>
> > it is not clear to me that the DPG approach (baseline or not) is highly effective. The constraints in the EBM are NOT satisfied in either
>
> The concern here is about the effectiveness of the whole DM line of work for controlled text generation with respect to constraint satisfaction rate,
>
> - **Full constraint satisfaction is trivial when disregarding “Catastrophic forgetting”**:
> If the deviation from the original language model is not controlled, full constraint satisfaction would become a trivial matter. For example for “topic control” a policy that is peaked on a few fluent sequences that satisfy the constraints would easily achieve 100% constraint satisfaction and very low perplexity. But such a policy would be useless as a language model as it has lost all its generalization capacities.  Therefore it is crucial to do distributional evaluation using the metrics in the paper such as $KL(\pi, a)$ and $KL(p, \pi)$ for which our GDC++ method excels in all experiments.
> Those arguments and an extra analysis on full constraint satisfaction are already done in (khalifa et al. 21 Appendix C) and similar insights on catastrophic forgetting are discussed before in many controlled text generation work (Datharthi et al. 21, Welbel et al. 21).
>
> - **Only DM approaches can handle distributional constraints:**  Distributional constraints have a large potential of solving the timely problem of bias in language models. Those constraints constitute a large portion of our experiments (Tasks 7-10).
>
> - **DM approaches perform on par with SOTA controlled text generation methods**
> The Distributional Matching approaches (DPG with and without baselines) are able to increase constraints satisfaction rate from < 1% on average to around 60%, this means that 60% of the generated sequences will contain the imposed constraints (e.g. a specific topic) compared to <1% of samples from the original language.
> This is on par with SOTA controlled text generation methods such as PPLM (Dathathi-20) and CTRL (Keskar-19) which achieve around ~50% constraint satisfaction for word list constraints (For reference see a comparison with DPG in (Khalifa-2021)).
>
>
> > While baselines improve the DPG objective, constraint satisfaction, as shown by the second subfigure in figure 3, is NOT improved,
>
> We present the superiority of GDC++ (with baseline) over GDC more clearly in an **additional table** in the appendix Table 5 page 25. Overall, GDC++ is better than GDC in 7/10 tasks in terms of constraint satisfaction.
>
> Additionally, as shown in Figure 4 the addition of baselines allows training with smaller batch sizes and the constraint satisfaction improvement between GDC++ and GDC is clear and evident in smaller batch sizes, e.g. 256.
>
> Moreover, the same observation on the superiority of GDC++ is displayed in our additional experiments on code-generation please see the comment **Extra Experiments on Code Generation**
>
> > Based on Algorithm 1, it seems that the DPG importance weights are not normalized, Is this the case?
>
> Note here that in the DPG case the normalization factor is a global multiplicative constant across all batches (equal to $Z$) that could be estimated globally (not with respect to each batch individually) and is absorbed into the learning rate of Adam. Empirically, in our early experiments, we didn’t find any difference between both implementations.
>
> Overall, we thank you again for your constructive feedback, we hope to have answered your concerns about the theoretical novelty and the efficiency of DM approaches, and we are hopeful that you might reconsider your overall assessment of our work.

---

> > ### Comment · Reviewer_Up1T · 2021-11-29
> > **Post-rebuttal comments**
> >
> > Thank you to the authors for their responses, and additional results.
> >
> > - Regarding novelty of baselines in DM, I maintain my position that adding a baseline to DM is a rather straightforward exercise, and lower in novelty (c.f. eq. 14 and 15).
> >
> > - Certainly a tradeoff in terms of deviation from the prior vs. constraint satisfaction is expected, but note that 1) The Ziegler approach generally satisfies the constraints better both in the DM paper (Khalifa et al., 2021), and in your new results (page 25) without degenerating like basic REINFORCE, and 2) Distribution matching can be retrofitted into the Ziegler approach via sampling the of the target constraint distribution. Considering this, a systematic comparison of the the tradeoff between prior and constraint satisfaction to understand how well the operating point can be controlled is still missing from the DM literature. Such analysis is needed and would inform future research efforts, and while somewhat orthogonal to the new baseline that is contributed, increase the contribution of the paper signficantly.
> >
> > - With the above said, the introduced baseline does lower variance, and lead to better optimization of the DM objective, leading to less deviation from the base LM for the same constraint satisfaction performance (the claims in the paper still need to be corrected to clearly reflect this reality). The authors have also provided additional results for the existing experiments (page 25) and results on code generation, which show a smaller but consistent advantage to utilizing the derived baseline, as expected.
> >
> > Revised assessment:
> >
> > - In consideration of the improvements to the paper and my remaining concerns about lower novelty and significance, I have increased my score from 3 to 5.

---

### Official Review · Reviewer_Yzso · 2021-11-07

**Correctness:** 4
**Technical Novelty And Significance:** 3
**Empirical Novelty And Significance:** 3
**Recommendation:** 6
**Confidence:** 4

**Main Review:**

* Strength
    * This paper proposes an interesting idea and interpretation to connect RM and DM paradigm
    * This paper proposes a variance reduction method for DPG which demonstrates its improvement on performance, stability and sample efficiency
* Weakness
    * From my understanding, the baseline mostly comes from the observation in 3.3, which has limited technical novelty. It will be great if authors can justify more on the technical novelty.
    * It will be cool to show some results on BLEU improvement on machine translation, or user studies on language generation tasks, to demonstrate its practical impact, as the quantitative metrics right now are still kind of artificial.


**Summary Of The Paper:**

This paper untangles the connections between the RM and DM paradigm, and exploit these connections to propose a baseline for the DPG method. Experiments on a set of controllable language generation experiments show that the propose baseline can lead to better performance, stability and sample efficiency.


**Summary Of The Review:**

This paper addresses an interesting direction to connect RM and DM paradigms and shows promising results for better controlled language generation with their proposed variance reduction technique. Though I'm not sure if the technical novelty is sufficient and the empirical results are strong or practical enough given limited technical novelty.

---

> ### Author Response · Authors · 2021-11-22
> **Reply to Reviewer Yzso**
>
> > This paper proposes an interesting idea and interpretation to connect RM and DM paradigm
> > a variance reduction method for DPG which demonstrates its improvement on performance, stability, and sample efficiency
>
> Thank you for highlighting some of the main contributions of our work and for showing your appreciation of them.
>
>
>
> ### Technical novelty
>
> > From my understanding, the baseline mostly comes from the observation in 3.3, which has limited technical novelty. It will be great if authors can justify more on the technical novelty.
>
> We acknowledge that an important part of the technical novelty of our work comes from a number of theoretical observations that are brought forward over several sections and it might not be easy to sort them out. For this reason, we provide in a separate thread an enumeration of the theoretical innovations in this paper, motivating them accordingly. **Please see our separate comment regarding “Theoretical Novelty’’**.
>
> ### Additional Experiments
>
> > It will be cool to show some results on BLEU improvement on machine translation, or user studies on language generation tasks, to demonstrate its practical impact, as the quantitative metrics right now are still kind of artificial.
>
> Thanks for the suggestion that motivated us to indeed think of extra tasks to demonstrate practical impact. We run extra experiments on code auto-completion using Language Modeling, a task that recently gained lots of attention and led to widely adopted tools such as Github Copilot, OpenAI Codex, and Tabnine.
>
> We fine-tune language models to generate python code that is compilable (a task that the pre-trained language models on code don’t satisfy most of the time). We evaluate the tasks using extra evaluation metrics that demonstrate the practical usability and quality of the generated code’s AST node count, length, and compilability rate. We find consistent gains by using our proposed method. **Please refer to the separate comment “Extra Experiments on Code Generation” for more details.**
>
>
>
> Overall, we appreciate your constructive feedback that has motivated us to elaborate our theoretical contributions and run extra experiments on code generation. We hope our rebuttal was convincing. If this is the case, we would appreciate it if you would increase your evaluation of our paper.

---

> > ### Comment · Reviewer_Yzso · 2021-11-30
> > **Response to Rebuttal**
> >
> > Thanks so much for addressing my comments! While I think the revised version with additional experiments strengthens empirical results, I still maintain my previous thought that the technical novelty is somewhat limited. I'll keep my score.

---

### Official Review · Reviewer_ifjN · 2021-11-08

**Correctness:** 3
**Technical Novelty And Significance:** 2
**Empirical Novelty And Significance:** 2
**Recommendation:** 5
**Confidence:** 4

**Main Review:**

While I think it's nice to analyze the connection between RM and DM, the math provided in the paper is simple, and the main contribution is just to add a baseline to the algorithm of Khalifa et al. I encourage the authors to try out ideas they mentioned as future work to have a substantial contribution for a conference publication.

Typos: In Sec 5.2 Tasks (f), "$\phi_2(x)=1$" should be "$\bar \mu_2=1$". In Sec 5.5 Gradient Variance, $\pi$ is missing in "$G_\theta(x)=A(x)\nabla_\theta \log_{\theta}(x)$"

**Summary Of The Paper:**

This paper shows a connection between the reward maximization (RM) approach and the distribution matching (DM) approach for fine-tuning language models. The authors further suggest borrowing the baseline idea from reinforcement learning and applying it to DM to reduce variance. Experiments show that adding a baseline not only makes training more stable, but also converges to a better solution.

**Summary Of The Review:**

I think this paper is below the acceptance threshold because it is just a simple addition to the method of Khalifa et al.

---

> ### Author Response · Authors · 2021-11-22
> **Reply to Reviewer ifjN**
>
> > While I think it's nice to analyze the connection between RM and DM, the math provided in the paper is simple, and the main contribution is just to add a baseline to the algorithm of Khalifa et al.
>
> Thank you for highlighting the importance of the main argument brought forward by our paper, namely establishing a connection between two seemingly distinct approaches: DM and RM.  We would like to reflect on the two criticisms raised in your review:
>
> - **The simplicity of the math:** Previous work failed to notice the connections we are making between DM and RM. While we show a simple derivation relating the two concepts, we argue that our observations are novel and far from trivial.
>
> - **Reducing our contribution to the addition of the baseline:** Our paper brings forward several theoretical contributions in addition to the theoretical and empirical work directly improving the GDC algorithm. For convenience, we enumerate these contributions more clearly in a separate paragraph **Please see our separate comment above: “Theoretical Novelty’’**.
>
>
> > Typos: In Sec 5.2 Tasks (f) ...
>
> Thanks for your thorough reading, we corrected those in the updated version of the paper.
>
>
>
> Overall, we hope our reply has addressed your concerns about the theoretical novelty in the paper. Please let us know if you have any further concerns or comments.

---

### Author Response · Authors · 2021-11-22
**We updated the Manuscript with Extra Experiments on Code Generation**

**For a more detailed view see the updated manuscript Appendix C page 21**

Based on some reviewers’ suggestions we designed and ran extra experiments on additional NLG tasks. In particular *code generation* using large language models, a problem that is gaining a lot of traction in the language modeling community as demonstrated in recent models such as OpenAI Codex (Chen et al. 2021).

The task we focused on was controlling a language model trained on Python code to only generate code that compiles using a python interpreter. We finetuned GPT-2 for python code and implemented the scorer $b(x)$ as a function that calls the Python interpreter to check whether running a string $x$ raises an exception.

Our additional experiments are included in the updated version of the manuscript (Appendix C page 21). Consistently with previous experiments, adding a baseline (GDC++) improves over GDC in terms of constraint satisfaction (the fraction of samples from the model that compiles) and KL divergences from the original language model $a$ and the target optimal distribution $p$. These improvements translate into better downstream metrics such as Self-BLEU-5.


|       | Compilability Rate (↑) | KL(p,π) (↓) | KL(π,a) (↓) | Self-BLEU5 (↓) |
|-------|------------------------|-------------|-------------|----------------|
| GDC   | 0.68                   | 0.48        | 0.15        | 0.89           |
| GDC++ | **0.69**                   | **0.46**        | **0.13**        | **0.88**           |

---

### Author Response · Authors · 2021-11-22
**Theoretical Contributions**

In response to the reviewers' request, we make loud and clear the theoretical novelty brought forward by this paper:

- **Introducing Parametric Rewards:** This novel generalization of standard RL that we introduce is a crucial step to link Distribution Matching and Reward Maximization. We are not aware of previous RL approaches making use of Parametric Rewards. Here, we present an in-depth treatment of the Policy Gradients algorithm generalizing to such rewards.


- **Baselines for Parametric Rewards:** a posteriori the simple form of baseline for DPGon ($B=Z$) might deceive the reader as looking like *a straightforward mathematical exercise* akin to the case of standard non-parametric rewards. We stress here that the underlying theoretical work is far from trivial. One cannot use this mean reward baseline out of the box and assume its correctness. Because “parametric rewards” depend on the policy’s parameters $\theta$, **it is not a priori evident whether subtracting the expected reward would bias the gradients nor whether the resulting advantage would have a zero mean**. Additionally, it is not evident how to extend that baseline form to the DPGoff case $\(B_{off}(x) = Z \frac{\pi(x)}{q(x)}\)$, without the principled derivation that we introduce in the paper and in the appendix.


- **Interpreting KL-Control under the lens of Distribution Matching:** We furthermore provide original insights about KL-control, reinterpreting it under the lens of DM. These insights sparked hours of discussions which even included the authors of (Ziegler et al.  19) themselves. Beyond understandability purposes, interpreting KL-control as an instance of DM has the potential of informing future work, such as, for instance, capitalizing on the arsenal of Reinforcement Learning algorithms for Distributional Matching objectives.

---

### Decision · Program_Chairs · 2022-01-20

**Decision:**

Reject

**Comment:**

This paper explores the connections between reward maximization (RM) with REINFORCE and distribution matching (DM) with distributional policy gradients (DPG) for fine-tuning language models. Based on this, the paper proposes to apply a baseline (an idea in reinforcement learning) in DM to reduce variance and improve sample efficiency. Reviewers have concerns on the technical novelty as claimed in the paper, since the application of baseline is a straightforward practice and the resulting method is a simple addition to the existing method. More analysis (such as on the tradeoff between prior and constraint satisfaction, etc) was also suggested.